# Mir-193b Regulates the Differentiation, Proliferation, and Apoptosis of Bovine Adipose Cells by Targeting the *ACSS2/AKT* Axis

**DOI:** 10.3390/ani10081265

**Published:** 2020-07-24

**Authors:** Zihong Kang, Sihuang Zhang, Enhui Jiang, Fachun Wan, Xianyong Lan, Mei Liu

**Affiliations:** 1Animal Nutritional Genome and Germplasm Innovation Research Center, College of Animal Science and Technology, Hunan Agricultural University, Changsha 410128, China; amaranthkang@163.com (Z.K.); wanfc@sina.com (F.W.); 2College of Animal Science and Technology, Northwest A&F University, Yangling 712100, China; sihuanzhang1990@nwafu.edu.cn (S.Z.); jiangenhui@yeah.net (E.J.)

**Keywords:** bta-miR-193b, *ACSS2*, differentiation, proliferation, apoptosis

## Abstract

**Simple Summary:**

Several studies have shown that miR-193b plays an important role in preadipocyte differentiation. Herein, we explored the role of bta-miR-193b in adipocyte development, using EdU, flow cytometry, CCK-8, RT-qPCR, Western blotting, and oil red O staining. The result showed that bta-miR-193b could regulate the proliferation, differentiation and apoptosis of adipocytes. The dual-fluorescent reporter vector experiments showed that bta-miR-193b directly targeted ACSS2, and the regulatory function of *ACSS2* is opposite to that of miR-193b. Meanwhile, we demonstrated that *ACSS2* could significantly promote the expression of AKT and pAKT proteins. In conclusion, our research provides new insights that confirm that bta-miR-193b inhibits bovine adipose cell proliferation, and promotes apoptosis by negative regulation of the *ACSS2*/AKT pathway.

**Abstract:**

The precise functions and molecular mechanisms of microRNAs (miRNAs) in adipocytes are primarily unknown. Studies have demonstrated that miR-193b plays a pivotal role in the differentiation of preadipocytes. Herein, we evaluated the effects of bta-miR-193b on the growth and development of adipocytes, using the EdU cell proliferation method, flow cytometry analysis, CCK-8 assay, RT-qPCR, Western blotting, and oil red O staining. We observed that the overexpression of bta-miR-193b significantly affected the differentiation, proliferation, and apoptosis of adipocytes. The results of the dual-fluorescent reporter vector experiments demonstrated that bta-miR-193b directly targeted Acyl-CoA synthetase short-chain family member 2 (*ACSS2*). Additionally, the effects of *ACSS2* overexpression on the proliferation and apoptosis in adipose cells were the opposite of those induced by bta-miR-193b. We also demonstrated that *ACSS2* can significantly promote the expression of AKT and pAKT proteins. Therefore, this study presents a novel mechanism by which bta-miR-193b regulates adipocyte development by targeting *ACSS2*.

## 1. Introduction

Adipose, one of the largest tissues in an animal body, is formed in the womb during the perinatal period and continues to grow throughout life. Remarkably, even in adults, new fat cells are produced at a constant rate [1]. However, despite the regulation of the metabolic balance by adipose tissue via the secretion adipokines, excessive adipose accumulation can cause metabolic disorders, leading to obesity [2]. Previous studies have demonstrated that excessive fat deposition is associated with the incidence of cancer, hypertension, type 2 diabetes mellitus, and other diseases [3]. At the cellular level, increasing the number and volume of fat cells leads to clinical syndrome [4]. Previous studies have demonstrated that many miRNAs can regulate the growth and development of mammalian adipocytes, and can have a potential effect on adipogenic differentiation disorders [5]. Therefore, elucidating the regulatory mechanisms of the growth and development of fat cells has turned out to be an important research direction for the prevention and treatment of obesity and associated metabolic diseases.

MicroRNAs (miRNAs) are highly conserved small endogenous non-coding RNAs (~18–25 nucleotides), that function as negative regulators of gene expression at the post-transcriptional stage [6]. Most miRNAs play vital roles in regulating cell proliferation, differentiation, apoptosis, and other biological processes [7]. Recent studies have demonstrated that miRNAs play an important role in adipogenesis [8,9]. For example, miR-204 targets sirtuin 1 (*SIRT1*) to promote adipogenesis, while miR-93 regulates obesity by inhibiting sirtuin 7 (*Sirt7*) and T-box 3 (Tbx3) [10].

Sun et al. observed that miR-193b is essential for the differentiation of primary brown adipose tissue (BAT) cells, and the expression of uncoupling protein 1 (UCP1) [11]. In addition, many studies have examined the regulatory mechanisms of miR-193b in other biological processes. For instance, miR-193b controls the production of adiponectin via pathways, involving nuclear transcription factor Yα and, possibly, nuclear receptor-interacting protein 1 [12]. In human gliomas, miR-193b can reduce the expression of SMAD family member 3 (Smad3) to promote cell proliferation [13]. Overexpression of miR-193b activates caspase 3/7, leading to the apoptosis of A2780 cells [14]. However, the functional mechanism of miR-193b and its target genes in bovine adipocyte differentiation, proliferation, and apoptosis remains unclear. Considering the regulatory model of miRNAs, RNAhybrid and TargetScan were used to screen and predict the potential target genes of bta-miR-193b. Acyl-CoA synthetase short-chain family member 2 (ACSS2) garnered interest after being predicted by bioinformatics analysis. Furthermore, ACSS2 was determined to be involved in the synthesis of fatty acids and sterols from free acetic acid [15]. In mice and humans, ACSS2 was confirmed as a target gene of sterol regulatory element-binding protein (SREBP) [16]. ACSS2 has been demonstrated to be transcriptionally regulated by SREBP-1 in goats [17]. However, no reports establishing the role of white adipose tissue in cattle have been published. The purpose of this study was to elucidate the mechanism of bta-miR-193b in the regulation of adipocyte growth and development.

## 2. Material and Methods

### 2.1. Sample Collection and RNA Extraction

Tissue samples were obtained from the fetal (*n* = 3), calf (*n* = 3), and adult cattle (*n* = 3), from the spleen, liver, fat, muscle, kidneys, heart, and lungs. The samples used in this study were obtained from a slaughterhouse in Xi’an, Shaanxi province. The subcutaneous adipose tissue was sectioned, immediately placed in Dulbecco’s modified Eagle’s medium (DMEM) at 37 °C, and transported to the laboratory for primary adipocytes separation [18]. 

### 2.2. Isolation and Culture of Bovine Adipose Cells

Adipocytes were isolated from bovine subcutaneous adipose tissue, in accordance with the protocol established in previous studies [18]. The cells were seeded in a medium containing 20% fetal bovine serum (FBS) with 1% antibiotics, and were maintained in a incubator with 5% CO_2_ at 37 °C. Adipocytes were harvested on 0, 2, 4, 6, 8, and 10 d after the induction of differentiation. 

### 2.3. Plasmid Transfection

The bta-pre-miR-193b sequence was obtained by PCR amplification, and then released via Kpn I and Xho I digestion. The bta-pre-miR-193b fragment was ligated into the pcDNA3.1(+) vector, using T4 DNA ligase. The CDS region of *ACSS2* was connected to the pcDNA3.1(+) vector, namely pcDNA-ACSS2. The ACSS2-3′-UTR fragment was amplified by PCR amplification, including the bta-miR-193b-binding site. Meanwhile, the ACSS2-3′-UTR mutant sequence was also amplified. The seed sequence was randomly mutated to prevent the binding of bta-miR-193b, and the mutation sequence was designed with the reverse primer. Specifically, a 5-base mutagen in the bta-miR-193b-binding site of ACSS2 3′UTR (ACSS2-3′-UTR mutant) was generated using a pair of mutagenic primers: 5′- CCCTCGAGTCGCTCAGAATGACCACG-3′ and 5′-ATTTGCGGCCGCCAACAAGCCTCAGCACTCCTGGA-3′. The fast enzymes, *Not*I and *Xho*I, were used for connecting two fragments to the psi-CHECK-2 vector, followed by ligation by T4 DNA ligase. The primer sequences are given in Appendix A.

### 2.4. Real-Time Quantitative PCR (RT-qPCR) and Western Blot Analysis

Total RNA from the tissues and cells was extracted using the TRIzol reagent, in accordance with the manufacturer’s instructions (TaKaRa, Dalian, China). The OD measurements were performed using a nucleic acid quantitative instrument, followed by the determination of the concentration of total RNA. The integrity of RNA was examined, and the presence of contamination was detected using 1% agarose gel electrophoresis. RNA was reverse-transcribed using a reverse transcription kit (TaKaRa Biotech Co. Ltd.), in accordance with the manufacturer’s instructions. Real-time quantitative PCR (RT-qPCR) was performed using the SYBR green detection dye and the CFX96 real-time qPCR detection system, in accordance with the protocol reported by Jin et al. [9] The expression of bta-miR-193b was normalized to that of U6. The expression of *ACSS2* was normalized to that of *GAPDH*. Primer information is given in Appendix A. The medium in the cell culture dish was aspirated and discarded. The cells were washed with PBS, digested with trypsin, and collected into a 1.5 mL centrifuge tube. The supernatant was collected after centrifugation. RIPA cell lysate containing PMSF protease inhibitor was added in a 4:1 ratio (Solarbio, Beijing). The protein was mixed with sodium dodecyl sulfate (SDS) loading buffer, and boiled for 10 min in a 98 °C water bath. The samples were added to SDS-polyacrylamide gel for separation by electrophoresis and then transferred to a 0.2-μm polyvinylidene difluoride (PVDF) membrane. Following electrophoresis, PVDF membranes were removed, soaked, washed with TBST solution for 10 min, and sealed with a sealant for 2 h, followed by washing again with TBST solution, three times for 10 min each. The washed PVDF membranes were incubated with primary antibody overnight, followed by three washes with TBST. Subsequently, the PVDF membrane was incubated with secondary antibody at room temperature for 2 h. Finally, the antibody-reacting bands were exposed using an ECL kit (Solarbio) [19]. The relevant antibody information is listed in Appendix A. Herein, the differentiation markers included mRNA expression levels of PPARG, CEBPA, and *FABP4* genes, as well as protein expression levels of PPARG and CEBPA. The proliferation markers included mRNA expression levels of *CyclinD*, *CyclinE,* and *PCNA* genes, as well as protein expression levels of CyclinD and PCNA. The apoptosis markers included mRNA expression levels of *Bax*, *Bcl2,* and *Caspase9* genes, as well as protein expression levels of Bcl2. 

### 2.5. Adipogenic Differentiation and Oil Red O Staining

When cell confluence reached 100%, the medium was replaced with induced differentiation medium. After two days, the cells were incubated with the differentiation medium, and the medium was changed every two days [18]. After adipocyte differentiation, the medium was discarded. The cells were washed with PBS, followed by the addition of 4% paraformaldehyde. The cells were placed at room temperature for 30 min. Subsequently, the 4% paraformaldehyde solution was discarded, and the cells were washed three times, and stained for 1 h using a solution containing oil red O. Finally, the oil red O dye was immediately discarded, and cells were washed four times with PBS. Images were captured under a microscope [9]. The oil red O was then eluted from the stained cells with isopropanol, and quantified using a microplate reader at 510 nm.

### 2.6. Cell Proliferation Assay

The CCK-8 proliferation assay uses a water-soluble tetrazolium salt to quantify the number of live cells, by producing an orange formazan dye upon bio-reduction in the presence of an electron carrier. The preadipocytes were seeded into 96-well plates, and further cultured until the cell density was observed to be approximately 50%. Then, pcDNA-miR-193b or pcDNA-ACSS2 was transfected into the cells, and the cells were cultured for 24 h, with double complete medium. When the cell density reached approximately 75%, 10 μL of CCK-8 reagent was added to each well and cultured for 2–4 h (MultiSciences (Lianke) Biotech Co., Ltd.; Hangzhou, China). After shaking the wells gently, the optical density (OD) was measured immediately, at a wavelength of 450 nm. The measurements were used for statistical analysis. The EdU proliferation assay provides a sensitive and robust method, to detect and quantify cell proliferation in live mammalian cells using fluorescence microscopy. Preadipocytes were inoculated in a 48-well plate. Subsequently, pcDNA-miR-193b or pcDNA-ACSS2 was transfected into the cells, and cells were cultured for 24 h. EdU medium was added to each well, and the cells were incubated for 3 h before fixation. Finally, detection was performed using a microscope [19]. Flow cytometry analysis for the cell cycle assay was performed as follows: the preadipocytes were cultured in a 60 cm^2^ culture dish in 4 mL medium. After 24 h, adipocytes were washed in PBS and fixed in 70% ethanol (ethanol: PBS = 7:3), overnight at −20 °C. Finally, detection was performed using a cell cycle testing kit. 

### 2.7. Dual-Luciferase Reporter Assay

HEK293T cells were inoculated into 96-well plates. When the degree of cell fusion reached approximately 50%, pcDNA-miR-193b was co-transfected with the ACSS2 3′-UTR psi-CHECK-2 vector, and the cells were cultured in twice the volume of medium for 24 h. Next, the steps in the luciferase reporter assay kit instructions (Promega) were followed, to measure the luciferase activity.

### 2.8. Statistical Analysis

All data were analyzed using one-way ANOVA (three or more treatments) and *t*-test (to compare two treatment groups) in SPSS 24.0 (IBM Corp., Endicott, NY, USA). RT-qPCR analysis of cDNA from each sample was performed in triplicate, and the relative gene expression was normalized to the expression of U6 or *GAPDH*, using the 2^−ΔΔ*C*t^ method. When *p* < 0.05, the difference was considered statistically significant. Data are presented as mean ± SE.

## 3. Results

### 3.1. Dynamics of Bta-miR-193b Expression during Adipocyte Differentiation

The similarity analysis indicated that the miR-193b seed sequence is highly conserved among cows, mice, and chickens, which implies the functional conservation of miR-193b (Figure 1a). Notably, the changes in the expression levels of bta-miR-193b were detected on days 0, 2, 4, 6, 8, and 10, during the process of differentiation. Initially, an increase in the expression levels of bta-miR-193b was observed, followed by a decrease in expression. The expression was significantly higher on the second day of differentiation, compared to the expression on the following days (Figure 1b). Based on the above results, we hypothesized that bta-miR-193b might be associated with adipogenesis.

### 3.2. Bta-miR-193b Promotes Early Differentiation of Cattle Adipose Cells

In order to explore whether bta-miR-193b regulates the differentiation of adipose cells, we overexpressed bta-miR-193b and stained the cells with oil red O for 5 days and 10 days after inducing differentiation. The results showed that, compared with the control group, more lipid droplet formation was detected in the pcDNA-miR-193b-transfected adipocytes by oil red O staining on day 5 of differentiation. However, there was no significant difference on the tenth day of differentiation (Figure 2b,c,h,i). After transfection with pcDNA-miR-193b and on the fifth day after differentiation, the expression levels of *PPARG*, *CEBPA*, and *FABP4* mRNA increased. In addition, a co-instantaneous increase in the PPARG and CEBPA protein expression levels was detected on the fifth day of differentiation (Figure 2d–f). However, the mRNA and protein expression levels were not significantly different on the tenth day of differentiation (Figure 2j–l).

### 3.3. Mir-193b Inhibits Cattle Adipose Cell Proliferation and Promotes Adipose Cell Apoptosis

With the purpose of exploring the function of bta-miR-193b in adipose cell proliferation, EdU, flow cytometry, CCK-8, RT-qPCR, and Western blotting were used to detect the proliferation of bovine adipose cells. Overexpression efficiency detection showed that the bta-miR-193b expression level was significantly increased (Figure 3a). The EdU incorporation assay was used to detect cell proliferation, and the data showed that bta-miR-193b obviously decreased the mitotic activity of adipocytes (Figure 3b,c). The flow cytometry assay demonstrated that bta-miR-193b decreased the number of S-phase adipocytes (Figure 3d). In addition, the CCK-8 test showed that bta-miR-193b significantly decreased the cell proliferation coefficient (Figure 3e). After the transfection of pcDNA-miR-193b, we found that the proliferation-related markers, including *cyclinD*, *cyclinE,* and *PCNA* mRNA and protein levels, were significantly decreased (Figure 3f–h). Subsequently, the experiment focused on exploring the effect of bta-miR-193b on apoptosis. First, we detected the mRNA expression levels of apoptosis-related markers. The results showed that overexpression of bta-miR-193b significantly increased *Bax* expression (Figure 3i). Meanwhile, Western blot analysis indicated that the Bcl-2 protein level was decreased (Figure 3j–k). These data showed that bta-miR-193b suppresses adipose cell proliferation, and may promote adipose cell apoptosis.

### 3.4. Bta-miR-193b Targets ACSS2 and the Biological Characteristics of ACSS2

In order to examine how bta-miR-193b affects the proliferation, differentiation, and apoptosis of adipocytes, RNAhybrid and TargetScan were used to screen and predict the potential target genes of bta-miR-193b. *ACSS2* was selected, based on the high context++score percentile, sequence characteristics, and biological functions (Figure 4a). The results demonstrated a significant decrease in the luciferase activity when the *ACSS2* wild-type vector plasmids were co-transfected with bta-miR-193b. However, no significant change was observed in the luciferase activity value in the co-transfected mutant vector group (*ACSS2* mut-type vector plasmids+ bta-miR-193b) (Figure 4b). Overexpression of bta-miR-193b was induced, and the consequent changes in the *ACSS2* mRNA expression levels were detected in bovine adipocytes. The results indicated that the *ACSS2* mRNA expression decreased significantly upon bta-miR-193b overexpression (Figure 4c). The aforementioned results demonstrated that *ACSS2* is a potential target of bta-miR-193b.

To elucidate the function of *ACSS2*, we first measured its expression levels in different tissues obtained from the cattle at various stages. The analysis of the expression profiles demonstrated that the expression of *ACSS2* was highest in the adipose tissue, indicating that it plays an important role in the cattle adipose tissue (Figure 4d). *ACSS2* expression levels increased significantly after six days (Figure 4e). Subsequently, we compared the expression levels of bta-miR-193b and *ACSS2* at different stages of differentiation. The results suggested that all components followed opposite trends (Figure 4f).

### 3.5. ACSS2 Promotes Cattle Adipose Cell Differentiation

Next, we studied the regulatory role of *ACSS2* in adipose cell differentiation. Overexpression efficiency detection demonstrated that the expression level of *ACSS2* was significantly increased (Figure 5a,g). The adipocytes transfected with pcDNA-ACSS2 formed a higher number of lipid droplets on day five or ten of differentiation than those transfected with the control, as demonstrated by oil red O staining (Figure 5b,c,h,i). The mRNA expression of *PPARG*, *CEBPA*, and *FABP4* were increased on the fifth or tenth day of differentiation after the transfection of pcDNA-ACSS2 (Figure 5d,j). The protein expression levels were not significantly different on the fifth day of differentiation (Figure 5e,f). However, a similar significant increase in the PPARG and CEBPA protein levels were found on the tenth day of differentiation (Figure 5k,l).

### 3.6. ACSS2 Promotes Bovine Adipose Cell Proliferation and Inhibits Cattle Adipose Cell Apoptosis

Since bta-miR-193b can directly regulate the mRNA expression of the *ACSS2* gene, the question arises whether the *ACSS2* also affects the proliferation of adipose cells. Therefore, the *ACSS2* overexpression vector was transfected to examine the role of *ACCS2* in adipocyte proliferation and apoptosis (Figure 6a). The EdU assay demonstrated that the overexpression of *ACSS2* can lead to a significant increase in the number of EdU-positive cells (Figure 6b,c). *ACSS2* expression also increased the number of S phase cells, as demonstrated by flow cytometry analysis (Figure 6d). In addition, the CCK-8 test showed that *ACSS2* could noticeably increase the cell proliferation coefficient (Figure 6e). The relative *cyclinD*, *cyclinE,* and *PCNA* mRNA levels were remarkably increased after the transfection of pcDNA-ACSS2. We observed a similar increase in the cyclinD and PCNA protein levels (Figure 6f–h). We wanted to investigate whether *ACSS2* regulates cellular apoptosis. RT-qPCR and Western blotting experiments indicated that *ACSS2* significantly increased the mRNA and protein expression levels of *Bcl-2* (Figure 6i,k). Collectively, the above results indicate that *ACSS2* promotes the proliferation of bovine adipocytes, and inhibits their apoptosis.

### 3.7. ACSS2 Promotes the AKT Signaling Pathway

Previous studies have demonstrated that miR-193b and ACSS2 can affect the AKT signaling pathway and AKT phosphorylation [20,21]. Hence, we also investigated whether bta-miR-193b and *ACSS2* influence the AKT signaling pathway. The results demonstrated that the *ACSS2* increased the AKT and p-AKT protein levels after 24 h, and on day 10. However, no significant change was observed in the AKT and p-AKT protein levels following the overexpression of bta-miR-193b compared to the expression of the control during the proliferation period (24 h) and differentiation period (induced differentiation on day 10). However, notable differences were observed in AKT and p-AKT protein levels when *ACSS2* was overexpressed, compared with the control in the proliferation period (24 h) and differentiation period (induced differentiation on day 10) (Figure 7). Overall, these results indicate that bta-miR-193b regulates the proliferation and apoptosis of bovine adipose cells via the bta-miR-193b-ACSS2/AKT axis (Figure 8).

## 4. Discussion

In recent years, miRNAs, as important regulatory factors, have turned into research hotspots in many domains, such as tumor and metabolism studies. The expression of miRNA differs among various tissues, such as the pancreas, adipose tissue, and liver, that are associated with obesity [22]. The expression of miRNAs regulates the balance of energy and metabolism in the body by controlling various metabolic pathways [23]. Previous reports have suggested that certain miRNAs are associated with fat formation [24]. For instance, miR-146b is favorable for lipogenesis [25]. In contrast, miR-27a and miR-27b inhibit lipogenesis [26,27]. Furthermore, in the adipogenesis state, miR-199a-3p inhibits adipocyte differentiation and facilitates adipocyte proliferation [28]. In 2011, Sun et al. reported that miR-193b plays a significant role in brown fat differentiation [11]. However, in 2013, Feuermann et al. reported that miR-193b did not regulate the growth and activity of brown fat in mice [29]. Studies have demonstrated the involvement of many regulatory factors and signaling pathways that are common to the differentiation of both BAT and WAT. The changes observed in the expression of most miRNAs during the differentiation of WAT and BAT are similar [30]. Therefore, this study was conducted to explore the exact function and specific mechanism of miR-193b in cell cycle progression and adipogenesis differentiation.

To understand the function of bta-miR-193b, the expression dynamics of bta-miR-193b during adipocyte differentiation were determined by RT-qPCR. The bta-miR-193b expression demonstrated a trend of increased expression, followed by a decrease. In addition, the similarity analysis indicated that the miR-193b seed sequence is highly conserved among cows, mice, and chickens, which implies the functional conservation of miR-193b. Our data demonstrated that the overexpression of bta-miR-193b resulted in an increase in the expression levels of differentiation marker genes on the fifth day of differentiation, and an increase in lipid droplets. However, no significant difference was observed on the tenth day of differentiation. Thus, bta-miR-193b promotes the early differentiation of bovine adipocytes. Cell proliferation and apoptosis are important biological processes. Apoptosis is a specific form of cell death. In terms of tissue kinetics, apoptosis can be considered as a mechanism that counterbalances the effect of cell proliferation [31]. Certain studies have shown that certain cell cycle-related protein families and BCL-2 protein families can regulate cell proliferation and apoptosis, respectively [32,33]. Our data demonstrated that bta-miR-193b downregulated the expression levels of proliferation-related genes during the growth stage of bovine adipocytes, while the expression level of proapoptosis genes was increased. Typically, cell proliferation and apoptosis are regulated via a series of complex interactions between *CyclinD* and the *Bcl2* pathway [34]. Therefore, the aforementioned results indicate that the overexpression of bta-miR-193b inhibited the proliferation of adipocytes. In conclusion, the effect of bta-miR-193b on the proliferation and apoptosis of adipocytes may balance the role of cell function.

Typically, miRNAs can block mRNA expression by binding to the 3′ UTR [35]. Therefore, we predicted that bta-miR-193b could target *ACSS2i*, based on the results of RNAhybrid and TargetScan. We observed that bta-miR-193b inhibits *ACSS2* mRNA expression. The dual-luciferase assay demonstrated that bta-miR-193b significantly inhibits luciferase activity, indicating that *ACSS2* is a target gene of bta-miR-193b. Many reports indicate that the *ACSS2* gene is involved in the synthesis of fatty acids in bovine breast tissue and adipose tissue. It also promotes the storage and utilization of body fat, by selectively regulating genes related to lipid metabolism [36]. However, no reports on the function of *ACSS2* in adipocyte proliferation, differentiation, and apoptosis in cattle have been published.

To elucidate the function of *ACSS2*, we first measured the expression levels of different tissues in the cattle at different stages. The results indicated that the expression of *ACSS2* was highest in adipose tissue, suggesting that it plays an important role in the cattle adipose tissue. We also compared the expression levels of bta-miR-193b and *ACSS2* at various stages of differentiation; the results demonstrate an opposite trend in the expression of both. However, our data also demonstrated that the overexpression of *ACSS2* significantly upregulated the expression of *PPARG*, *CEBPA,* and *FABP4* genes at the mRNA level, as well as that of PPARG and CEBPA at the protein level. The formation of lipid droplets was also enhanced, as detected by oil red O staining, on the fifth and tenth day of differentiation. This indicates that bta-miR-193b primarily acts during early differentiation and the *ACSS2* primarily acts during late differentiation. Additionally, bta-miR-193b may not target *ACSS2* to promote adipocyte differentiation. Changes in their expression during adipocyte differentiation may also be regulated by other complex factors. We observed that, during the growth phase of adipocytes, *ACSS2* upregulated the expression of proliferation-related genes. The results of the EdU, flow cytometry, and CCK-8 assay were consistent, indicating that the overexpression of *ACSS2* promotes the proliferation of adipocytes. However, the expression of *Bax* decreased. In conclusion, our data indicate that *ACSS2* is an important target of bta-miR-193b in bovine adipose cell proliferation and apoptosis.

It has been reported that bta-miR-193b and ACSS2 can affect the AKT signaling pathway and phosphorylation of AKT [20,21]. Thus, we also investigated whether miR-193b and ACSS2 affected the AKT signaling pathway. It is noteworthy that the increase in the expression level of *ACSS2* leads to an increase in the expression of AKT and p-AKT proteins during the proliferation and differentiation periods. However, no significant changes were observed in the AKT and p-AKT protein levels when bta-miR-193b was overexpressed, compared with the overexpression of the control during the proliferation and differentiation periods. It can be inferred that the effect of *ACSS2* on adipocyte proliferation and differentiation may be realized through the modulation of the AKT signaling pathway.

## 5. Conclusions

In conclusion, the results of the present study confirm that bta-miR-193b inhibits bovine adipose cell proliferation and promotes apoptosis by the negative regulation of the *ACSS2*/AKT pathway. However, the specific molecular mechanism of miR-193b and its target gene *ACSS2* in adipocyte differentiation remain to be elucidated.

## Figures and Tables

**Figure 1 animals-10-01265-f001:**
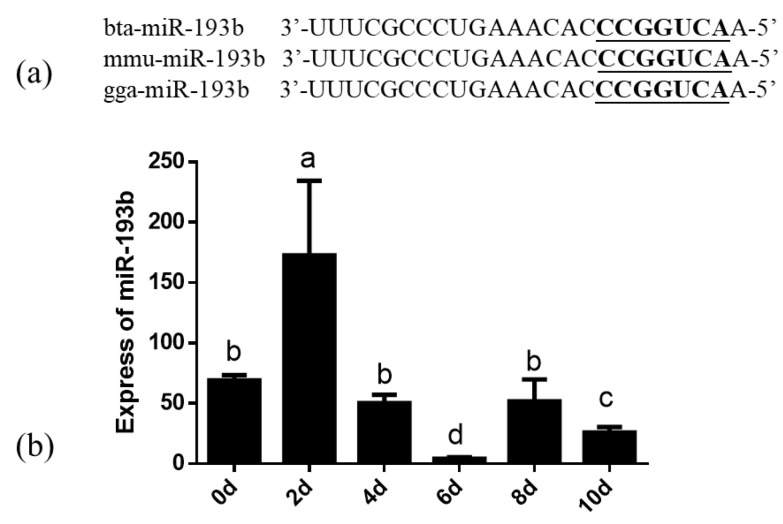
(**a**) The similarity analysis of miR-193b in cattle, mouse and chicken. The underlined font refers to the sequence of the seed. (**b**) The expression dynamics of bta-miR-193b during adipocyte differentiation determined by RT-qPCR. Different letters (a, b, c and d) represent significant differences as follows: *p* < 0.05.

**Figure 2 animals-10-01265-f002:**
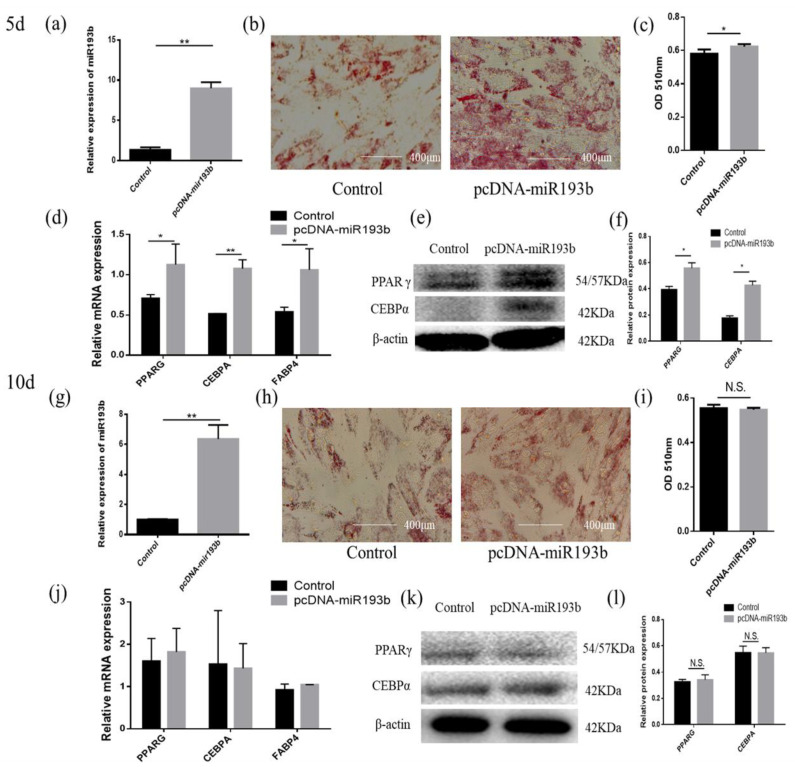
Bta-miR-193b promotes adipose cells’ early differentiation. (**a**,**g**) Overexpression efficiency detection at day 5 d or 10 d. (**b**,**h**) Oil Red O staining of adipose cells with bta-miR-193b at day 5 d or 10 d. (**c**,**i**) The results quantified by microplate reader at 510 nm. (**d**–**f**,**j**–**l**) Overexpression of bta-miR-193b in adipose cells and continues induced to undergo differentiation for 5 or 10 days, and subsequent detection the mRNA expression of *PPARG*, *C/EBPA* and *FABP4* by RT-qPCR compared with control group, and differentiation of adipose cells detected by western blot. The expression of miRNAs was normalized to U6. Values are mean ± SD for three biological replicates, * *p* < 0.05, ** *p* < 0.01; N.S. represent non-significant.

**Figure 3 animals-10-01265-f003:**
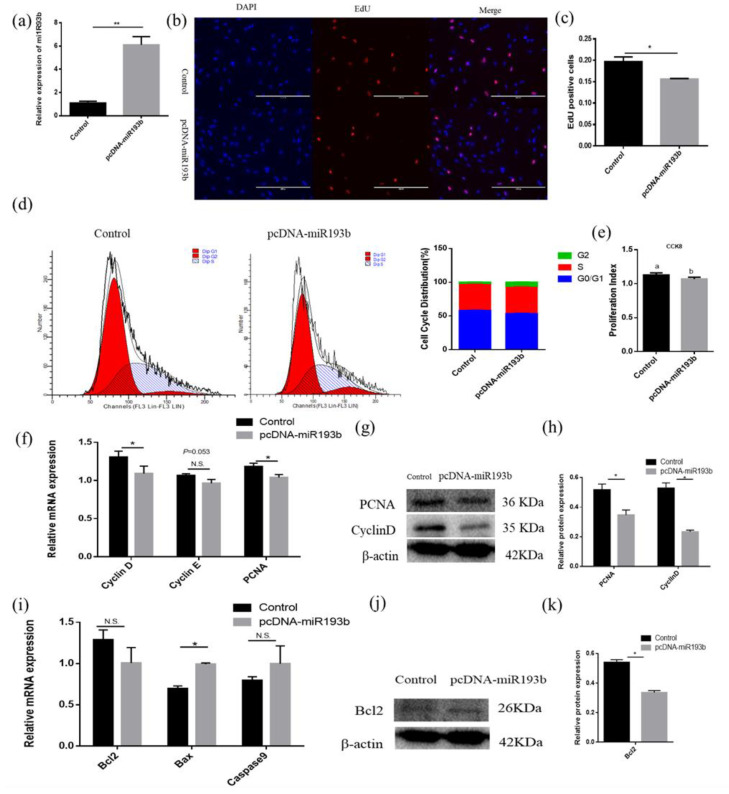
Bta-miR-193b inhibits adipose cells proliferation and promotes adipose cells apoptosis. (**a**) Overexpression efficiency detection. (**b**) EdU detected cell proliferation index; the scale bar stand 200 μm. DAPI (blue), EdU (red). (**c**) Analysis results of EdU positive cells. (**c**) Cell cycle distribution was detected by PI flow cytometry after a gain of bta-miR-193b. (**d**) Cell proliferation status was detected at 450 nm with CCK-8 reagent, after gain of bta-miR-193b. (**e**) Cell cycle-related genes (*cyclinD*, *cyclinE* and *PCNA*) mRNA expression level was detected by RT-qPCR after a gain of bta-miR-193b. (**f**–**k**) Expression of cell cycle-related and apoptosis-related genes were detected by RT-qPCR and Western blot. Values are mean ± SD for three biological replicates. Different letters (a, b) represent significant differences as follows: *p* < 0.05; * *p* < 0.05, ** *p* < 0.01; N.S. represent non-significant.

**Figure 4 animals-10-01265-f004:**
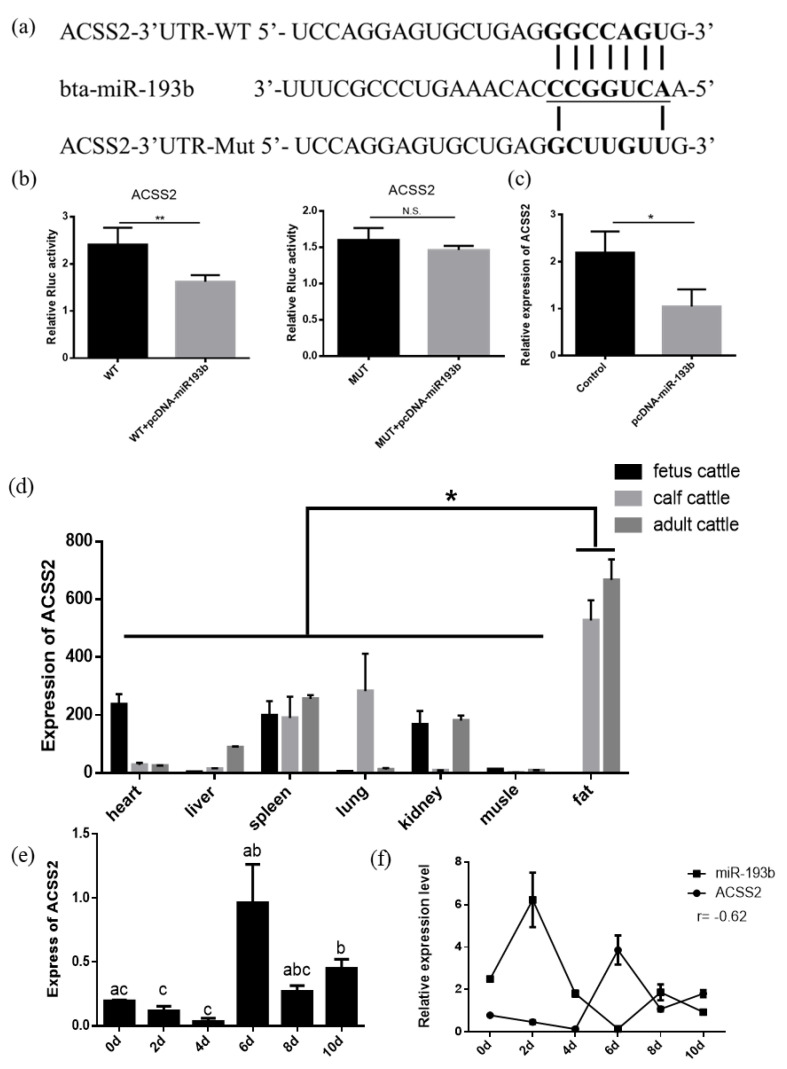
*ACSS2* is a target gene of bta-miR-193b. (**a**) The predicted target sites between bta-miR-193b and *ACSS2* 3’UTR. (**b**) The results of Dual luciferase reporter assay of potential target of bta-miR-193b. (**c**) The relative expression of *ACSS2* mRNA in bovine adipose cells was detected by RT-qPCR after a gain of bta-miR-193b. (**d**) The expression profile of *ACSS2*, in fetal, calf, and adult bovines. (**e**) The expression dynamics of *ACSS2* during adipocyte differentiation determined by RT-qPCR. (**f**) The expression characteristic of bta-miR-193b and *ACSS2*, in the differentiation stages of bovine adipose cells. Different letters (a, b, and c) represent significant differences as follows: *p* < 0.05; * *p* < 0.05; N.S. represent non-significant.

**Figure 5 animals-10-01265-f005:**
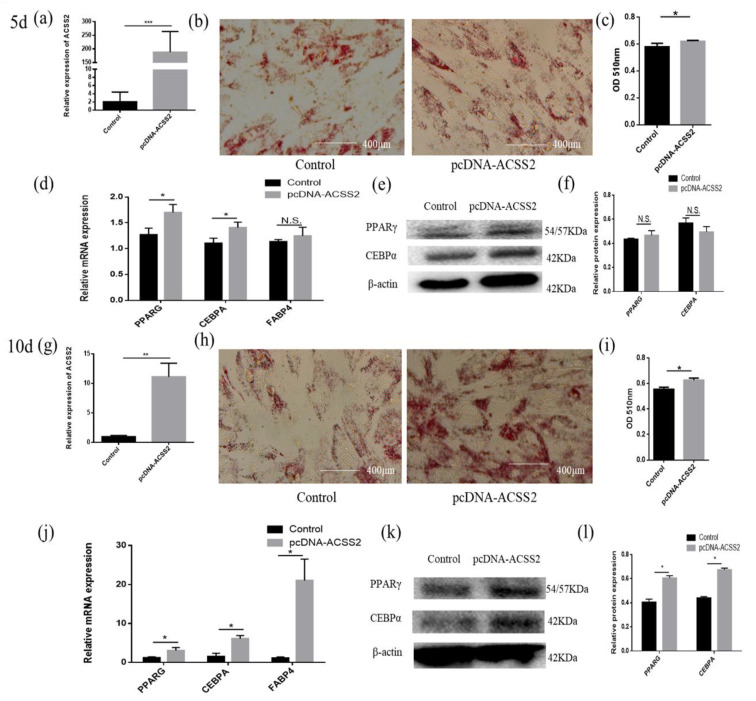
ACSS2 promotes adipose cells differentiation. (**a**,**g**) Overexpression efficiency detection at day 5 d or 10 d. (**b**,**h**) Oil Red O staining of adipose cells with bta-miR-193b at day 5 d or 10 d. (**c**,**i**) The results quantified by microplate reader at 510 nm. (**d**–**f**,**j**–**l**) Overexpression of ACSS2 in adipose cells and continues induced to undergo differentiation for 5 or 10 days, and subsequent detection the mRNA expression of PPARG, C/EBPA and FABP4 by RT-qPCR compared with control group, and differentiation of adipose cells detected by Western blot. The expression of miRNAs was normalized to U6. Values are mean ± SD for three biological replicates, * *p* < 0.05, ** *p* < 0.01, *** *p* < 0.001; N.S. represent non-significant.

**Figure 6 animals-10-01265-f006:**
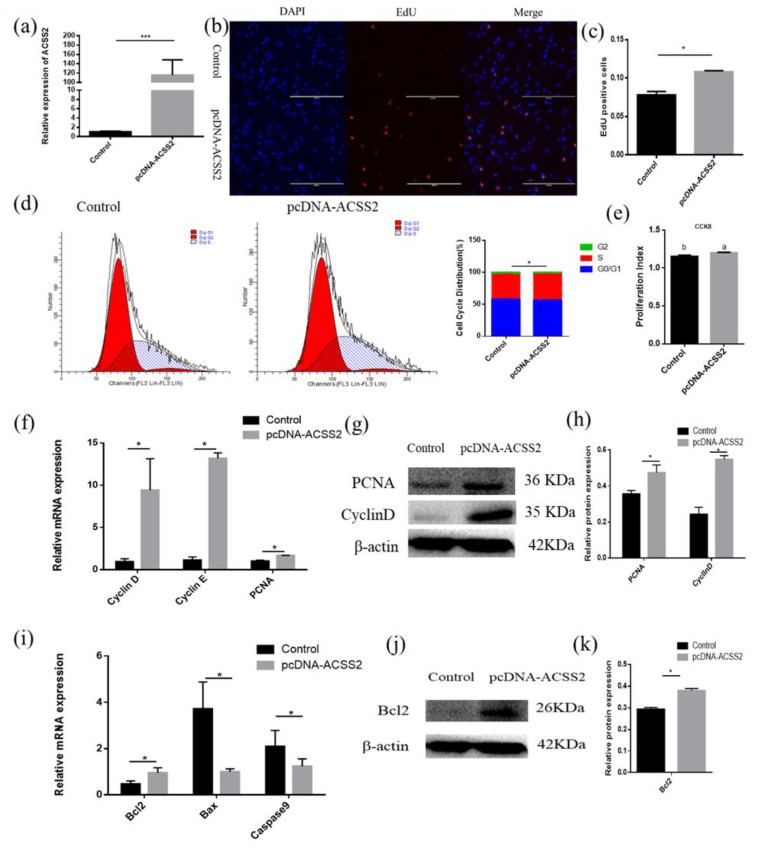
*ACSS2* promotes adipose cells proliferation and inhibits adipose cells apoptosis. (**a**) Overexpression efficiency detection. (**b**) EdU detected cell proliferation index, the scale bar stand 200 μm. DAPI (blue), EdU (red). (**c**) Analysis results of EdU positive cells. (**c**) Cell cycle distribution was detected by PI flow cytometry after a gain of *ACSS2*. (**d**) Cell proliferation status was detected at 450 nm with CCK-8 reagent after gain of *ACSS2*. (**e**) Cell cycle-related genes (*cyclinD*, *cyclinE* and *PCNA*) mRNA expression level was detected by RT-qPCR after a gain of *ACSS2*. (**f**–**k**) Expression of cell cycle-related and apoptosis-related genes were detected by RT-qPCR and western blot. Values are mean ± SD for three biological replicates. Different letters (a, b) represent significant differences as follows: *p* < 0.05; * *p* < 0.05, *** *p* < 0.001.

**Figure 7 animals-10-01265-f007:**
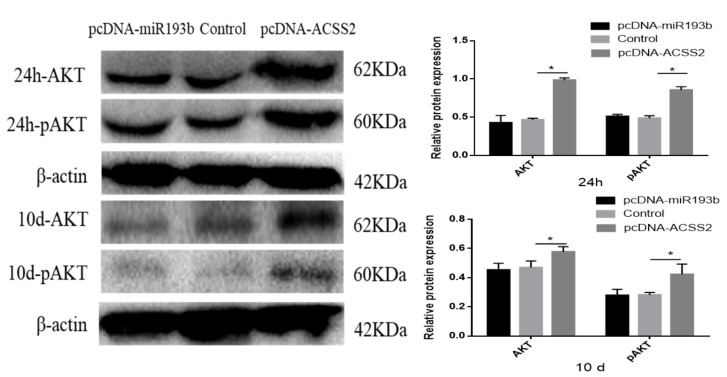
Bovine adipose cells transfected with pcDNA-miR-193b or pcDNA-ACSS2, AKT/pAKT proteins were detected by Western blot analysis. Values are mean ± SD for three biological replicates. * *p* < 0.05

**Figure 8 animals-10-01265-f008:**
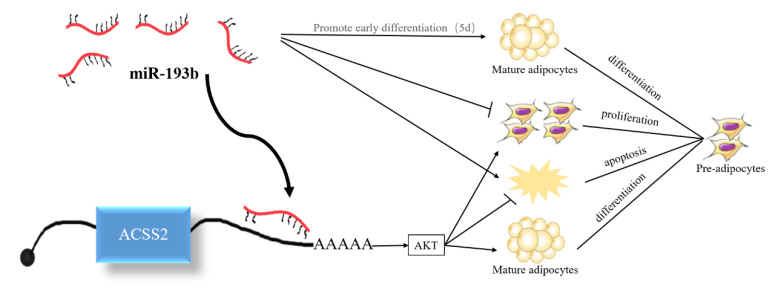
Proposed model of bta-miR-193b regulation on adipocyte growth and development by targeting *ACSS2*.

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
