# Peer review of "Mir-193b Regulates the Differentiation, Proliferation, and Apoptosis of Bovine Adipose Cells by Targeting the ACSS2/AKT Axis"

_animals, 2020, doi:10.3390/ani10081265_

Round 1

Reviewer 1 Report

The manuscript deals with the relationship between a miRNA and a gene in adipose cells. The work is of interest as the chosen miRNA function is nowadays not fully understood and its behaviour with candidate genes is of interest in general and specifically in the bovine species.

Authors use transfection of mutated products of both miRNA and a candidate gene to detect the expression of their transcripts and to validate the differentiation of the adipose cells. The work is of interest but I have some comments which should be taken into account before published.

Firstly, authors should introduce some more information on the bioinformatics analysis which directs them to the Acyl-CoA synthetase short-chain family member 66 2 (ACSS2) as candidate gene and no other important genes implicated in fat metabolism. Also, authors should explain in the method part, what are the mutations they introduce in the vector for miR193b and ACSS2 and why these sequences.

All over the text, figures are too complex and difficult to follow. The legend must be more concise (e.g. what mean the figures on the right of Figure 2 and others, with no numbering?), Too many data in the figures, it is rather difficult to read. Material and Method is both too precise and too short to understand what is being done, I would revise it explaining what is CCK-8 assay or EdU, a short explanation would be informative.  The whole text has a lot of details but it should be more concise explaining exactly what is needed to understand the different steps: introduction of mutations to invalidate the miRNA and the ACSS2 transcript, detection of expression with different tools etc.

Some other comments:

Abstract: First and third sentences are redundant, modify the text. Better explain the tools used

In Material and methods section, for the transfection protocol explain which is miR193b and ACSS2-3′-UTR mutant sequences.  

Figure 1 legends  lacks information: what means the underlined sequence?

The bta-miR-193b expression changes should be analysed to detect significant results and such as in line 146 not stating “The expression showed an increasing trend and then decreased; the highest expression level was reached after 2 d of differentiation … “

Figure 2 is not clear: what means the figures on the right with no numbering? Too many data, it is rather difficult to read.

Figure 3: part b cannot be seen. And on the right side again no explanation. On the Yaxis for Edu positive cells does this mean that as miR193b is transfected and increases expression, the number of adypocite cells is declining?

Line 205 “highscore” of what?

Line 209 here authors should explain what is the mutant vector group as no indication is given before in the text

Figure 4 again is not clear. Part b specifically is not clear at all.

Better explain all the steps you have done. For example line 248, I would explain that overexpression of ACSS2 has been performed through transfection???

Line 276 authors must say how they investigated if AKT signalling pathway is implicated. No introduction for this, the election of this pathway must be explained.

Discussion should be more concise, e.g. line 308 “Cell proliferation and apoptosis are important processes in fat development” Why? Discuss please. Line 313-316 Discuss why these specific genes have been studied.  Line 332 I do not understand this part of the sentence “and lipid drops (Oil Red O staining) on the fifth or tenth day of differentiation”.

Reviewer 2 Report

Kang and colleagues have analysed the role of miR-193b in cell cycle progression and adipogenesis differentiation in catle. The have overexpressed this miRNA in adipocytes and analysed the consequences in adipocyte differentiation, proliferation and apoptosis. Their results show that this miRNA is overexpressed during the first steps of differentiation, inhibits cell proliferation and promotes apoptosis. Moreover, the authors have analysed ACSS2 as a target of this miRNA. They have demonstrated the role of miR-193-b in ACSS regulation and shown opposite expression of both genes during adipogenesis. Contrary to miR193-b, ACSS overexpression promotes proliferation and inhibits apoptosis, however, it also promotes adipogenesis and increases the expression of adipogenic markers. The authors conclude that ACSS2 is a target of miR-193b in proliferation and apoptosis regulation but not in adipogenesis.

I think the manuscript is well written and it is easy to follow. The relevance of the study has been correctly explained and the methodology used is adequate to contrast their hypothesis. Results also support conclusions and I think it deserves publication. I only have minor comments and suggestions that could improve the manuscript, mainly related to clarify the methodology used.

Methods:

  • Please, explain the method you use to quantify miR-193b during adipogenesis. Did you use spike-in? How did you normalize its expression?
  • Explain droplet quantification using Oil Red O staining
  • How did you quantify ASSC2 in the different tissues, explain the units of y axis in Figure 4c. Methodology for this quantification is not included in the section.
  • Similarly, explain ACSS2 expression in figure 4d and 4 e. I understand you quantified the expression during adipogenic differentiation but it is not clear in the text.

Results:

  • There is a mistake in line 258. ACSS2 not miR-93 significantly increased Bcl2 mRNA expression.

Others:

  • Explain the meaning of EdU and CCK8 in the abstract and in the text.
  • Specify Cell-cycle testing kit in 2.6 cell proliferation assay

Reviewer 3 Report

The following manuscript discusses the role of miR-193b in the adipocyte differentiation. The results of the manuscript are nice and could be of interest for the topic of the research. However, the manuscript is not well organized in many sections. Introduction is too generic. Authors are invited to improve the literature and link the role of the miR-193b to the context by making hypothesis too. In Materials and methods, details of the procedures are missing. They are sometimes reported in the figure legends. Discussion should be improved as well. In this section, hypotheses are missing as well as a good analysis of the literature. Overall, english writing is difficult to be understood and many sentences are not really clear. Therefore, the manuscript needs major revisions before decision for publication can be done.

Comments and questions

Line 138 Please indicate when you used  one way Anova and t-test to compare what?

Line 95 Change “Quantitative PCR” to “Real Time Quantitative PCR (RT-qPCR)”, use it for all the manuscript

Line 125 What EdU assay is for?

Line 174 Flow cytometry is not indicated in the Materials and Methods. Which was the antibody used?

Lines 213-214 “Many reports indicate that the ACSS2 gene….” This sentence is more for discussion section

Line  331 PPARG, CEBPA, and FABP4, you should indicate this info in the materials and methods as well and in the context of the results

Line 317- 324 ACSS2 is reported to be a gene related to the storage and utilization. Could you please explain why using literature and link it to your results with hypothesis?

Overall figure legends are not clear. They report details that can be used in the materials and methods.

Figure 1b it is not clear here which was the parameter used for evaluating gene expression. Have you used Delta Ct or what? Please indicate

Figure 2. The figure legend is not well explained. The figures on the right top of are not included in the text

Figure 7 To understand if there is a change of protein expression using western blotting raw data is really tricky with these figures. This is also for figures 5e and 5f, 6f and 6h. I would recommend to analyze the intensity of the band in order to make graph bars as it is normally done for gene expression

Figure 8 it is a nice picture but it should be explained more

Reviewer 4 Report

In the manuscript entitled "miR‐193b regulates differentiation, proliferation, and apoptosis of bovine adipose cells by targeting the ACSS2/AKT axis" the Authors set out to identify the role of miR-193b in cattle adipocyte development. In general the manuscript provides interesting results. However the low number of samples used and the lack of description of quality and purity of RNA samples raise some concerns. The hypothesis is clear and the methods adequate. The conclusion goes a little too far for the obtained results and should be reviewed. Moreover the manuscript must be read by an English native speaker. Below are the specific comments.

line 20-21 - the sentence is unclear, please rephrase.

Please provide full name of genes abbreviations' once used for the first time in the text.

line 74 - I found the lack of consistency in conclusions description, I do not think it is necessary to mention abut beef cattle; or please indicated more about it in the text and other summaries parts in the manuscript.

How biological sample used for RNA extraction were protect from degradation. Moreover, how quality of those samples was determined. How results obtained form qRT-PCR were evaluated?

In my opinion there is too much discussion in "Results section" that should be corrected.

Round 2

Reviewer 3 Report

The authors have fully answered to my recommendation. Therefore, the manuscript can be accepted for publication at this time.